# IQA-EVAL: Automatic Evaluation of Human-Model Interactive Question Answering

**Ruosen Li**[1], **Ruochen Li**[1], **Barry Wang**[*2], and **Xinya Du**[1]

[1]Department of Computer Science, University of Texas at Dallas
[2]Department of Computer Science, Carnegie Mellon University
[1]{ruosen.li, ruochen.li, xinya.du}@utdallas.edu
[2]barryw@cs.cmu.edu

## Abstract

To evaluate Large Language Models (LLMs) for question answering (QA), traditional methods typically focus on assessing single-turn responses to given questions. However, this approach doesn't capture the dynamic nature of human-AI interactions, where humans actively seek information through conversation.[2]. Recent works in human-computer interaction (HCI) have employed human evaluators to conduct interactions and evaluations, but they are often prohibitively expensive and time-consuming to scale. We introduce an automatic evaluation framework IQA-EVAL to achieve Interactive Question Answering Evaluations[3], more specifically, we introduce a LLM-based Evaluation Agent (LEA) that can: (1) simulate human behaviors to generate interactions with IQA models; (2) automatically evaluate the generated interactions. Moreover, we propose assigning personas to LEAs to better simulate groups of real human evaluators. We show that: (1) our evaluation framework with GPT-4 (or Claude) as the backbone model achieves a high correlation with human evaluations on the IQA task; (2) assigning personas to LEA to better represent the crowd further significantly improves correlations. Finally, we use our automatic metric to evaluate five recent representative LLMs with over 1000 questions from complex and ambiguous question answering tasks, which comes with a substantial cost of \$5k if evaluated by humans.

## 1 Introduction

The advent of Large Language Models (LLMs) has significantly advanced the field of natural language processing (NLP), enabling systems to perform a wide range of tasks with remarkable proficiency [Zhao et al., 2023; Wei et al., 2022a; Yang et al., 2024; Du, 2024b; Jing et al., 2024]. Among these tasks, question answering (QA) has emerged as a critical and representative goal-oriented application, demonstrating the potential of LLMs to generate informative responses as an assistant [Biancofiore et al., 2024]. Multiple methods have been proposed to enhance the faithfulness and explainability of generated information [Wei et al., 2022b; Long, 2023; Li and Du, 2023; Du, 2024a]. Beyond developing these methods, rigorous evaluation of the generated outputs is also crucial.

Accurate and consistent evaluation helps researchers understand existing LLM's capacities and emerging human-LLM QA interactions [Chang et al., 2023; Lin and Chen, 2023; Chang et al., 2023]. Traditionally, automatic metrics such as accuracy have been used to evaluate models based on the

---

[*]Work done while at the Department of Computer Science, Cornell University.
[2]Details are in Appendix F
[3]https://github.com/du-nlp-lab/IQA-Eval

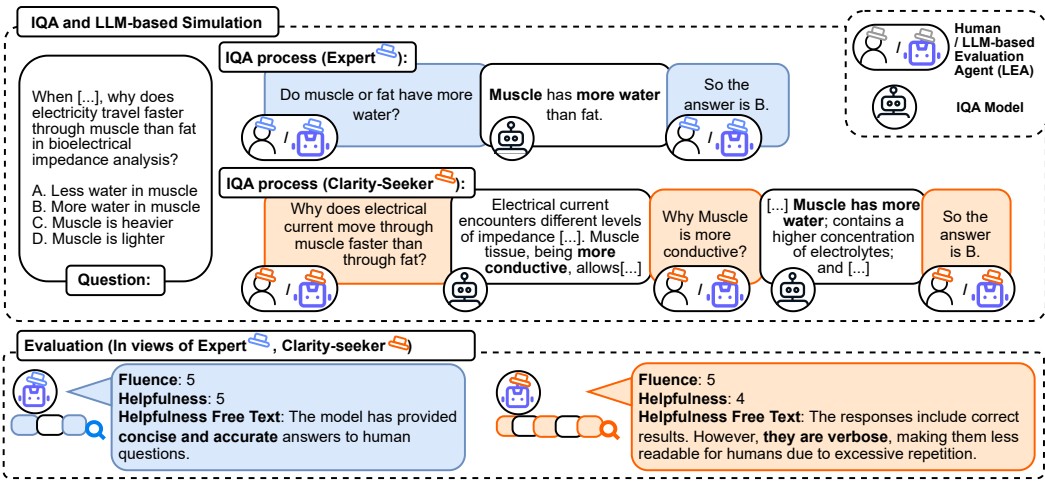

Figure 1: An example of human-model interactive question answering (IQA) and our automatic evaluation (IQA-EVAL). The two interactions occur with two types of personas in humans (or LLM-based evaluation agents): **Expert** and **Clarity-seeker**, and are evaluated by humans or agents with corresponding personas. The IQA model only responds to the immediately preceding prompt without further contexts like the question itself (leftmost in the Figure).

*quality* of their direct answers to specific questions. However, as *interactions* between humans and LLMs grow more complex and nuanced, these traditional metrics often fail to capture the full spectrum of a model's capabilities (e.g. helpfulness and fluency), particularly in interactive QA settings [Liu et al., 2016; Deriu et al., 2021], whereas interactions are crucial for user experience and system effectiveness, yet remains overlooked in traditional evaluation paradigms. Recent works such as [Lee et al., 2023] evaluated human-LLM interactions, a process that involves human participation and annotation. Although human evaluations for these interactions provide a closer approximation to real-world use cases, this approach is significantly costly and time-consuming. Recognizing the need for automatic evaluation, works like G-Eval [Liu et al., 2023b], LLM-Eval [Lin and Chen, 2023], FaithScore [Jing et al., 2023], and PRD [Li et al., 2024] proposed to automate the assessment of non-interactive LLM responses using LLMs as evaluators.

Drawing insights from (a) using LLM for automatic evaluation and (b) literature of LLM-agents research Wang et al. [2024]; Deshpande et al. [2023], we propose IQA-EVAL framework to auto-evaluate the performance of IQA models (i.e. LLMs) with LLM-based Evaluation Agent (LEA) to simulate and then evaluate interactions. By additionally incorporating personas, our experiments on a well-annotated dataset show that our methods align well with human judgments and provide a more comprehensive evaluation of LLMs in interactive settings than traditional metrics. Finally, we benchmark recent LLMs with new complex and ambiguous questions, and demonstrate that the accuracy of answers does not always transfer to the corresponding ranking of models on their capability of achieving good human-model interactions. See an overview of our system in Figure 1.

Our contributions are as follows:

- We propose the first LLM agent-based automatic evaluation framework IQA-EVAL designed specifically to generate and then evaluate interactions in IQA. Our results demonstrate a strong correlation with human evaluations.
- We propose persona-based LLM evaluation agents to better assess how models adapt to different user preferences and interaction styles.
- We experiment with IQA-EVAL framework to benchmark the most recent LLMs on the IQA task, demonstrating the strength and general effectiveness of our framework on fully automated evaluation.

## 2 Related Work

**Evaluating Interactions** Traditional methods for human-model dialogue evaluation have often been centered around single-turn pairwise evaluation [Vinyals and Le, 2015; Li et al., 2016]. Some

methods with multi-turn Likert scores emerge [Venkatesh et al., 2017; Zhang et al., 2018; See et al., 2019] but require time-consuming and costly collection of human-model conversations. In response, Ghandeharioun et al. [2019] suggests a self-play scenario where a dialog system engages in conversation with itself, employing a set of proxy measures for evaluation.

Acute-eval [Li et al., 2019] and LLM-Eval [Lin and Chen, 2023] have advanced multi-turn evaluation frameworks, reflecting the increasing demand for sophisticated techniques that holistically capture human-AI interactions. Further studies in specific interaction domains like task completion [Liu et al., 2023a], code generation [Yang et al., 2023], and collaborative problem-solving [Lee et al., 2023; Huang et al., 2023; Fu et al., 2023] emphasize the need for evaluations that consider both environmental and human elements [Wang et al., 2023b]. Our work differs from these methods by introducing an automated approach that emphasizes interaction quality and significantly reduces the reliance on human annotations.

**LLM-based Agent for Simulation**   Recently, LLMs rises to demonstrate human-like intelligence, as evidenced in various studies [Xiao et al., 2023; Rao et al., 2023; Li et al., 2023; Jiang et al., 2023; bench authors, 2023; Brown et al., 2020; Touvron et al., 2023; Chowdhery et al., 2022]. The integration of LLMs into agents that simulate complex human behaviors and social interactions is an area of growing research interest [Maes, 1995; Wooldridge and Jennings, 1995; Xi et al., 2023]. For example, Park et al. [2022] and Gao et al. [2023] employ these agents to simulate human emotions, attitudes, and behaviors in social networks, while Park et al. [2023] leverages in-context learning to simulate human behaviors in sandbox world. Moreover, Argyle et al. [2022] utilizes "algorithmic bias" inherent in GPT-3 to reflect the response patterns of different human subgroups. Horton [2023] utilize LLMs in experimental setups of behavioral economics experiments to facilitate pilot studies. Additionally, Hämäläinen et al. [2023] and Wang et al. [2023a] investigate LLM-based agents in recommender systems to simulate and collect data on user behavior. These studies show the broad applicability and potential of LLM-based agents in simulating human behaviors and interactions across diverse applications. Our work utilizes LLM-based evaluation agents (LEAs) to fully automate interactive quality assessments, handling both interaction generation and evaluation, to enhance the evaluation of IQA models in realistic scenarios.

**Personas in NLP**   Personas are constructed profile prompts that represent key traits of a group of users, as defined in the HCI field, reflecting their characteristics, behaviors, and goals to guide the design of technologies that are well-suited to user needs [Cooper et al., 2014]. This approach enhances relevance and personalization in NLP applications [Nargund et al., 2022; Bamman, 2015; Sheng et al., 2021; Zhong et al., 2020], offering significant potential for customizing engagement and improving the effectiveness of conversational agents [Li et al., 2016; Zhang et al., 2018; Chan et al., 2019; Madotto et al., 2019; Zheng et al., 2019]. Li et al. [2016] introduces a persona-based neural conversation model to enhance dialogue personalization and coherence. Zhang et al. [2018] develops personalized dialogue agents that incorporate user-specific details to enhance interaction. In our work, persona settings enable our framework to tailor interactions and assessments, aligning more closely with the specific characteristics and preferences of different user groups.

## 3   IQA-EVAL: Evaluating Interactive Question Answering (IQA)

In this section, we introduce our IQA-EVAL framework for automatically evaluating Interaction Question Answering Models (IQA models) with **LLM-based Evaluation Agents (LEA)**. LEAs are used to simulate humans in the following two stages: (1) generating interactions with IQA models; and (2) evaluating interactions. Lastly, we discuss the use of personas to for LEAs.

### 3.1   Interaction Generation with LEA (Stage 1)

Inspired by peer discussions Lee et al. [2023]; Wang et al. [2024], we prompt LEAs to simulate human behaviors for effective interaction generation with IQA models. The structured prompt includes three key components: (1) a role description; (2) a task description; and (3) instructions for the discussion.

**Role Description** outlines the people that the LEA model will simulate during interactions. For example, the description for a standard persona could be: `You are mimicking a human.`

**Task Description** briefly describes the action that LEA model needs to perform in the task. For example, in a multi-choice question answering task, the prompt could be structured as follows: `You`

are trying to choose the correct answer for the given question. Both role and task descriptions can be adjusted based on the persona, as discussed in Section 3.3.

**Discussion instruction** guides LEAs on their subsequent steps by providing detailed descriptions to facilitate progress in interactions. It comprises two essential components: (1) the actions to take; and (2) the detailed procedures to follow. For example, in a question answering task, the prompt specifies:
```
You can ask an assistant questions for help.  Please ask sub-questions
to approach answers.  In each turn, please only ask one sub-question to
interact with the assistant.

In the sub-question, please include all necessary information in the
original question, such as the question and all options.  If you know
the answer, please output "So, the answer is:  A, B, C, or D".
```

At the start of an interaction, the LEA receives a system prompt that includes all three components above, along with the specific question to be addressed. As the LEA interacts with the IQA model, it generates sub-questions to request clarification of unknown entities, definitions, or particular aspects of the original question. Then, the IQA Model takes the questions as the input and output responses. After receiving responses, the LEA continues to pose further questions until it determines the final answer. The full prompt structure and interaction details are provided in Appendix C.2.

## 3.2  Interaction Evaluation with LEA (Stage 2)

Inspired by G-eval [Liu et al., 2023b], which demonstrates that evaluations by GPT models align closely with human assessments in NLG tasks, we propose utilizing LEAs for interaction evaluation. LEAs assess interactions generated by LEAs and IQAs in Stage 1. The module takes task details, such as questions or articles, and interactions as the input, and output evaluation scores. The prompt contains three parts: (1) role and task description; (2) metrics definition; and (3) evaluation instruction.

**Role and task description** instructs LEA to conduct evaluation. The role description acts the same as the role description in the previous stage. Moreover, it briefly describes the evaluation task. The general prompt looks like: `You are a helpful and precise evaluator who checks the quality of the AI assistant's responses in interactions.`

**Metrics definitions** describes the criterion that LEA needs to follow in the evaluation process. They can customized for different tasks. For the question answering task, we add the following prompt to define the "helpfulness" metric:
```
Helpfulness (5-point Likert):  How helpful was having access to the
AI Assistant compared to not having access?
```

Both the aforementioned parts can be tailored according to the persona that the LEA simulates in Stage 1, as detailed in Section 3.3.

**Evaluation instruction** outlines the specifics of the evaluation task and the required output format. This section may appear as a separate part of the prompt. For example, the instruction within the prompt might be structured as follows:
```
Please evaluate the above interactions between user and AI assistant
by using the following metrics:
<Metric definitions>
Please output each of the above metrics line-by-line.
```

Finally, all evaluation scores for metrics are calculated by averaging the results of multiple runs. The complete prompt is available in Appendix C.1, and further details about our implementation can be found in Section 4.

## 3.3  Assigning the Personas to LEA

Both aforementioned evaluation stages typically use a default persona. While this constitutes a somewhat neutral baseline in knowledge, language proficiency, and beliefs to bo baseline, individual users often exhibit diverse personal preferences and characteristics, making a one-size-fits-all evaluation less effective. Moreover, the persona distribution of the target user group significantly impacts the performance of IQA models in real-world applications. For instance, if 20% of human users prefer

brief interactions and 70% prefer detailed information, applying a general LEA to simulate this group of persons is likely to result in poor correlation with downstream users.

To better simulate the diversity of the groups of people and provide individualized evaluations, we assign personas to LEAs. This affects prompts of both interaction generation and interaction evaluation processes.

For example, when the LEA is assigned with the "Critical-Seeker" persona (definition in C.3) for interaction generation, we adapt the default role and task description (in Stage 1) to: `You prefer interactions rich in critical information. You need help from an assistant and try to get critical information from it to answer the following questions.` For interaction evaluation, the default role and task description prompt changes to `The AI Assistant should provide straightforward, simple, and concise answers to aid users in deducing solutions.` Additionally, the definition of metrics is also adjusted to align with this persona, with further details available in Appendix C.3.

## 4 Meta-Evaluation of IQA-EVAL Framework

To measure how our framework provides trustworthy IQA evaluations that align with human preferences, we conduct meta-evaluations experiments and report correlation scores.

### 4.1 Experiment Settings

**Dataset and Evaluation Metrics**    We apply our evaluation method on the annotated dataset from the study by Lee et al. [2023]. This dataset consists of 3641 interactions from 331 annotators. Questions in the dataset are multi-choice and are derived from the MMLU dataset [Hendrycks et al., 2020] (example question in Figure 1). The construction of MMLU requires each worker to participate in 11 random conversations with one of the following three IQA models: `TextDavinci` (`text-davinci-001`), `TextBabbage` (`text-babbage-001`), and `Davinci` (`davinci-001`). At the end of conversations, fluency and helpfulness scores are annotated by annotators. The number of queries and accuracy for each IQA model can be easily deduced from annotations. In this work, We adjust the four metrics to evaluate generated interactions:

- **Fluency (5-point Likert)**: How clear (or fluent) were the responses from the AI Assistant?
- **Helpfulness (5-point Likert)**: Independent of its fluency, how helpful was having access to the AI Assistant compared to not having access?
- **Number of Queries**: Counts the number of interaction turns in the conversation. This metric helps assess the efficiency of the AI in resolving queries within a minimal number of interactions.
- **Accuracy**: Quantifies how accurately the AI's responses match the golden answers. This is critical for evaluating the correctness of the AI's knowledge and its application in practical scenarios.

**LEA Models**    To evaluate the effectiveness of IQA-EVAL framework, we experiment with different LEA models on the above-mentioned three LLMs IQA models in MMLU. For LEA that conducts both interaction generation and interaction evaluation, we use `ChatGPT` (`GPT-3.5-turbo-1106`), `GPT4` (`GPT-4-1106-preview`), `Claude` (`Claude-1`).

**Evaluation of IQA-EVAL Framework**    We report **Pearson correlations** as the measrue of agreement between the Human evaluations and LEA evaluations of IQA models.

### 4.2 Experiment Results

According to Table 2, all models, including `GPT4`, `GPT3.5`, and `Claude`, show high correlation with human evaluations. `GPT4` aligns most closely with human judgments in both "Helpfulness" and "Fluency" metrics and the highest overall correlation score. This indicates that these models are capable of effectively performing IQA-EVAL framework as LEA models.

In Table 1, `ChatGPT` scores closest to human judgments, particularly in the "Helpfulness" metric. Conversely, `GPT4` and `Claude` score lower on "Helpfulness" than human evaluations, because they tend to produce inaccurate and repetitive responses that lack coherence and do not directly address

Table 1: IQA-EVAL evaluation results of IQA models (TDA: `TextDavinci`; TB: `TextBabbage`; DA: `Davinci`). **Bold numbers** indicate they are the most close to human results. The empty set symbol (Ø) indicates the number cannot be calculated due to the model's inability to follow instructions and produce a gradable answer.

| Evaluator | Helpfulness | | | Fluency | | | # Queries | | | Accuracy | | |
|---|---|---|---|---|---|---|---|---|---|---|---|---|
| | TDA | TB | DA | TDA | TB | DA | TDA | TB | DA | TDA | TB | DA |
| Human | 4.60 | 3.84 | 3.52 | 4.35 | 3.84 | 3.22 | 1.78 | 2.57 | 2.66 | 69.00 | 52.00 | 48.00 |
| IQA-EVAL-GPT4 | 3.67 | 2.30 | 2.10 | 4.77 | **3.87** | 3.03 | **1.57** | 2.27 | **2.37** | 0.87 | 0.83 | 0.67 |
| IQA-EVAL-Claude | 4.13 | 3.03 | 3.00 | **4.47** | 3.47 | **3.23** | 2.20 | **2.67** | 2.07 | **0.67** | **0.53** | 0.57 |
| IQA-EVAL-GPT3.5 | **4.30** | **3.87** | **3.93** | **4.47** | 3.67 | 3.97 | **1.57** | 1.77 | 2.00 | 0.63 | 0.47 | **0.53** |

user queries, as indicated by their generated explanations. IQA-Eval scores on "Fluency" are close and highly correlated to human judgments. Both scores given by humans and LEA models show that IQA models provide fluent outputs. Furthermore, according to the "# Queries" metric, most models conclude conversations more quickly than humans, except for `Claude`, which requires more turns, potentially due to its non-OpenAI origins that it needs more turns to adapt the conversational style and understand responses. Notably, `GPT4` achieves the highest accuracy among all models. Moreover, we consider the impact of self-enhancement bias and conduct more experiments. Details are in Section 6.4.

**Analysis of LEA for Stage 2 (Evaluating Interactions)**  For Stage 2 itself, we measure the LEA's capability of evaluating interactions, based on real human-generated interactions from Stage 1.

Table 2: Pearson Correlation ($\rho$) between IQA-EVAL evaluations and human judgments.

| | Helpfulness | Fluency | Overall |
|---|---|---|---|
| IQA-EVAL-GPT4 | **0.652** | **0.591** | **0.613** |
| IQA-EVAL-Claude | 0.640 | 0.552 | 0.551 |
| IQA-EVAL-GPT3.5 | 0.621 | 0.523 | 0.510 |

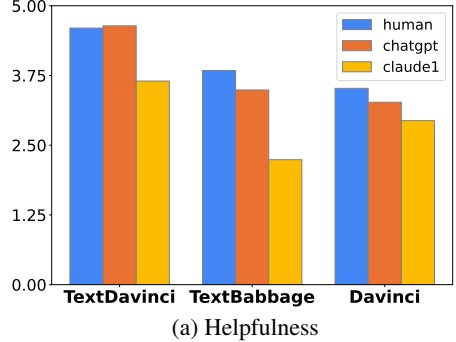

(a) Helpfulness

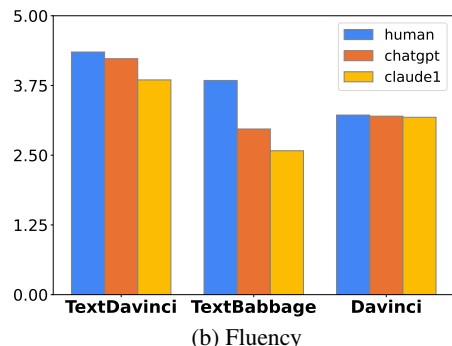

(b) Fluency

Figure 2: Interaction evaluation results evaluated by human and two LEA models on interactions between real human and IQA Models. All scores are on a scale of 5.

Results are in shown Figures 2a and 2b. The Pearson correlation coefficients for fluency and helpfulness between human judgments and LEA evaluations show distinct patterns. `ChatGPT` demonstrates a stronger correlation with human ratings, recording a correlation score of 0.424 for fluency and a correlation score of 0.306 for helpfulness. In contrast, `Claude` shows slightly lower correlations (0.281 for fluency and 0.287 for helpfulness). This shows that `ChatGPT` aligns better with human compared to `Claude` in these specific metrics.

Moreover, for interactions between humans and IQA models (Figure 2), LEA evaluations moderately correlate with human evaluations. However, for interactions between LEAs and IQA models (Table 2), which is the main focus of our paper, LEA evaluations highly correlate (around 0.6) with human evaluations. This indicates that LEA models are helpful when participating in the whole evaluation process, including both the interaction and evaluation. However, when evaluating interactions between humans and IQA models, LEAs focus differently from humans, which causes moderate correlations.

Thus, the results show that (1) both models' evaluations moderately correlate with human evaluation; (2) `ChatGPT`'s evaluation is closer and related to human evaluation than `Claude`'s.

### 4.3 Further Analysis for Free-form Feedback

Apart from the metrics above, we also prompt LLM-based Evaluation Agent (LEA) to explain the reason for generating scores in the free-form text format. We find that: 1) `ChatGPT` **Generates more human-like, positive reviews**: `ChatGPT` evaluations are generally more positive, frequently using terms like "helpful", "relevant", and "useful" – words not always noted by workers in their annotations. Despite this, `ChatGPT` often identifies similar issues as human raters, such as the provision of irrelevant and repetitive information. Overall, `ChatGPT`'s assessments align well with human evaluations; 2) `Claude` **flags more Issues**: `Claude` is more strict and critical to IQA models in interactions. In the free-text feedback, `Claude` tends to highlight more issues with model responses rather than acknowledging positive aspects, especially for `Davinci`. For one question, after interacting with the IQA model/assistant (`ChatGPT` in this case), human and two LEAs provide the following feedback:

```
Human:  When rephrasing questions well, the answers could be found in the
AI's response.
ChatGPT: The AI assistant was helpful in providing relevant information,
but there were issues with the accuracy.
Claude:  The AI assistant's responses were not very helpful.  The
responses were often vague, repetitive, or did not directly answer the
question.
```

## 5    Effect of Assigning Persona to LLM Evaluation Agent (LEA)

### 5.1    Persona Definitions

As discussed in Section 3.3, we assign personas to LEAs to simulate different groups of humans for diverse human alignments. We investigate assigning the following personas to LEAs, which are defined based on the crowdworker survey results in Lee et al. [2023].

- **Expert:** knowledgeable; quickly learns new concepts and applies them in the reasoning process to answer questions.
- **Critical-Thinker:** people who prefer critical information rather than redundant or detailed responses.
- **Adaptability-Seeker:** people who prefer assistants can understand their questions even if they are not precise.
- **Clarity-Seeker:** people who prefer clear explanations from assistants.

Based on the survey results on the distributions of personas of workers Lee et al. [2023], for each persona $P$, we split the crowdworkers into two groups: "persons with persona $P$", and "normal persons without specific persona $P$". For the first group, we initialize the role prompt in Section 3.3 and use it for the LEAs. For the second group, we utilize default prompting for LEAs. The LEA model in this section is ChatGPT(`GPT-3.5-turbo-1106`).

### 5.2    Experimental Results

In Table 3, we show the results of model (with personas) evaluations. To accurately simulate persona distribution, each interaction is executed multiple times, with different personas (including the standard one) assigned to LEA based on their distribution proportions. This method ensures that each persona's influence and characteristics are proportionally represented in the simulation, reflecting their respective prevalence within the overall distribution. The final score for a persona is an average of all experiment results.

The "Expert" persona decrease LEA query counts as "Expert" already possesses relevant knowledge and only needs key explanations. The "Clarity-Seeker" requires the most interaction turns among all personas for comprehension, but achieves the highest accuracy with `TextBabbage` and `Davinci` through detailed understanding of questions.

"Critic-Thinker" and "Adapability-Seeker" in Tables 3 and 4 rarely surpass upon the standard persona's human-preference alignment. We hypothesize that these personas are less reflected within the overall distributions of human preferences. In Table 3, the varying "# Queries" across personas reveals their significant influence on LEA interaction strategies. Accuracy remains consistent after adding personas, showing no performance degradation.

Together, these results indicate that assigning specific personas steer LEAs to perform IQA-EVAL in a more fine-grained and human-aligned way.

Table 3: IQA-EVAL evaluation results of IQA models (TDA: `TextDavinci`; TB: `TextBabbage`; DA: `Davinci`). LEAs, based on `GPT3.5`, are assigned specific personas when representing specific groups of workers.

| Evaluator | # Queries | | | Accuracy | | |
|---|---|---|---|---|---|---|
| | TDA | TB | TD | TDA | TB | TD |
| Human | 1.78 | 2.57 | 2.66 | 0.69 | 0.52 | 0.48 |
| IQA-EVAL | 1.57 | 1.77 | 2.00 | 0.63 | 0.47 | 0.53 |
| IQA-EVAL (Expert) | 1.20 | 1.49 | 2.20 | **0.73** | 0.56 | 0.53 |
| IQA-EVAL (Critical-Thinker) | 1.55 | 1.80 | 1.99 | 0.68 | 0.54 | 0.55 |
| IQA-EVAL (Adaptability-Seeker) | 1.50 | 1.75 | 2.10 | 0.66 | 0.52 | 0.55 |
| IQA-EVAL (Clarity-Seeker) | 1.64 | 2.10 | 2.34 | 0.63 | **0.57** | **0.57** |

Table 4: IQA-EVAL evaluation results (helpfulness and fluency) of IQA models. Correlations are between the LEA evaluation in each row and human evaluations.

| Evaluator | Helpfulness | | | | Fluency | | | | Overall |
|---|---|---|---|---|---|---|---|---|---|
| | TDA | TB | TD | $\rho$ | TDA | TB | TD | $\rho$ | $\rho$ |
| Human | 4.60 | 3.84 | 3.52 | - | 4.35 | 3.84 | 3.22 | - | - |
| IQA-EVAL | 4.30 (±0.06) | 3.87 (±0.11) | 3.93 (±0.13) | 0.621 | 4.47 (±0.05) | 3.67 (±0.08) | 3.97 (±0.06) | 0.523 | 0.510 |
| IQA-EVAL (Expert) | 4.17 (±0.08) | 3.08 (±0.09) | 3.12 (±0.11) | 0.756 | 4.47 (±0.02) | 3.84 (±0.04) | 3.40 (±0.04) | 0.787 | 0.670 |
| IQA-EVAL (Critical-Thinker) | 4.44 (±0.08) | 4.02 (±0.13) | 4.08 (±0.17) | 0.711 | 4.64 (±0.06) | 3.97 (±0.08) | 4.10 (±0.08) | 0.624 | 0.634 |
| IQA-EVAL (Adaptability-Seeker) | 4.24 (±0.05) | 3.67 (±0.11) | 3.75 (±0.11) | 0.713 | 4.52 (±0.08) | 3.84 (±0.07) | 3.84 (±0.09) | 0.637 | 0.650 |
| IQA-EVAL (Clarity-Seeker) | 4.45 (±0.07) | 3.77 (±0.15) | 3.80 (±0.12) | 0.747 | 4.60 (±0.04) | 3.85 (±0.04) | 3.94 (±0.06) | 0.676 | 0.690 |

It is worth nothing that our analysis of persona reassignment shows IQA-EVAL is sensitive to incorrect assignments (see Appendix E). Moreover, further analyses about bias evaluation, as well as measuring complementary metrics like offensiveness, are in Appendix G.

# 6 Benchmarking LLMs with IQA-EVAL on more Types of Questions

## 6.1 Datasets

To evaluate the robustness and generalizability of our evaluation framework, we conduct benchmarking across different models on two distinct question answering datasets, each offering unique challenges and complexities requiring advanced reasoning. **AmbigQA** Min et al. [2020] is a collection of 14,042 annotated questions sourced from the NQ-OPEN benchmarks Kwiatkowski et al. [2019a], an open-domain QA dataset. It focuses on questions with inherent ambiguities, reflecting the complexity encountered in real-world queries. These ambiguities often involve diverse aspects such as events, entity references, and answer types, resulting in multiple plausible answers for each question. **HotpotQA** Yang et al. [2018] comprises 113,000 question-answer pairs sourced from Wikipedia, which require multi-hop reasoning spanning multiple documents. It contains a rich array of intricate questions that demand the synthesis of information from various texts to determine accurate answers. In this benchmark, we select 500 questions from each dataset to form a dataset containing 1,000 complex multi-hop and ambiguous questions.

## 6.2 LLMs to Benchmark

Table 5: IQA-EVAL benchmarking results on HotpotQA and AmbigQA datasets.

| IQA Models | HotpotQA | | | | AmbigQA | | | |
|---|---|---|---|---|---|---|---|---|
| | Helpfulness ↑ | Fluency↑ | # Queries↓ | Accuracy↑ | Helpfulness | Fluency | # Queries | Accuracy |
| `TextDavinci` | 4.72 | 4.87 | 1.22 | 0.45 | - | - | - | - |
| `TextBabbage` | 4.70 | 4.88 | 1.74 | 0.37 | - | - | - | - |
| `Davinci` | 4.27 | 4.52 | 1.68 | 0.32 | - | - | - | - |
| `GPT3.5` | 4.72 | 4.95 | 1.49 | 0.63 | 4.91 | 4.97 | 1.89 | 0.60 |
| `GPT4` | 4.78 | 4.96 | 1.12 | 0.66 | 4.89 | 4.95 | 1.06 | 0.72 |
| `Claude` | 4.82 | 4.99 | 1.26 | 0.58 | 4.89 | 4.94 | 1.36 | 0.62 |
| `Llama2` | 4.70 | 4.95 | 1.32 | 0.55 | 4.96 | 4.94 | 1.79 | 0.52 |
| `Zephyr` | 4.64 | 4.88 | 1.01 | 0.40 | 4.38 | 4.66 | 1.03 | 0.45 |

Apart from `TextDavinci`, `TextBabbage` and `Davinci` we benchmark more LLMs: `GPT3.5`, `GPT4`, `Claude`, `Llama2` and `Zephyr`. The checkpoints for `Llama2` and `Zephyr` are Llama-2-7B and Zephyr-alpha, respectively. `GPT3.5` is used as LEA in our experiments.

Table 6: Comparison between new prompts and our prompts used in Table 1. The new prompts are more complex and include effective debiasing instructions.

| LEA models | Helpfulness | | | Fluency | | | Accuracy | | |
|---|---|---|---|---|---|---|---|---|---|
| | TDA | TB | DA | TDA | TB | DA | TDA | TB | DA |
| Human | 4.60 | 3.84 | 3.52 | 4.35 | 3.84 | 3.22 | 0.69 | 0.52 | 0.48 |
| IQA-EVAL-GPT4 (Our Prompts) | 3.67 | 2.30 | 2.10 | 4.77 | 3.87 | 3.03 | 0.87 | 0.83 | 0.67 |
| IQA-EVAL-GPT4 (New Prompts) | 3.50 | 2.23 | 2.10 | 4.40 | 4.07 | 3.53 | 0.87 | 0.83 | 0.67 |

Table 7: Comparison between new prompts and our prompts used in Table 5 on benchmarking LLMs with IQA-EVAL .

| LEA models | Helpfulness | Fluency | # Queries | Accuracy |
|---|---|---|---|---|
| IQA-EVAL-GPT3.5 (Our Prompts) | 4.72 | 4.95 | 1.49 | 0.63 |
| IQA-EVAL-GPT3.5 (New Prompts) | 4.68 | 4.91 | 1.35 | 0.60 |

## 6.3 Benchmarking Results

The IQA evaluation benchmarks are presented in Table 5. We divide IQA Models into two categories: weak IQA Models (`TextDavinci`, `TextBabbage`, and `Davinci`) and strong IQA Models (`GPT3.5`, `GPT4`, `Claude`, `Llama2`, and `Zephyr`). Weak IQA Models can assist the LEA with answering HotpotQA questions, but due to their knowledge limitations, they cannot help much with AmbigQA questions. `Zephyr` achieves the lowest performance compared to other strong IQA Models. On the HotpotQA dataset, `Zephyr`'s accuracy performance is only comparable to the strongest one among weak IQA Models, `TextDavinci`.

Most "Helpfulness" and "Fluency" scores are high (exceeding 3 out of 5), especially for strong IQA Models like `GPT4`. For the "# of queries", it is uncommon for interactions to extend beyond two turns. As on HotpotQA, most interactions conclude at the beginning of the second turn, as IQA models have effectively guided users to reach the answers. For AmbigQA, some conversations last longer whereas LEA spends additional turns on clarifying ambiguous entities before approaching the final answer. Additional benchmarking results on the Natural Question dataset are in Appendix D.

## 6.4 Self-Enhancement Bias

LLMs are shown to demonstrate self-favouring behaviours [Panickssery et al., 2024], and no verified or accessible mitigations to this issue exist to the best of our knowledge. This issue is particularly concerning when the LEA models evaluating the IQA models share the same underlying model. In this section, we discuss our two of our attempts to assess the effects of this bias.

Following Zheng et al. [2023] and Furniturewala et al. [2024], we included some empirically useful debiasing instructions as follows:

```
Please act as an impartial and unbiased judge.  In your evaluation,
please be objective and do not include any bias or your preference.
```

In Table 5, scores on the row of `GPT3.5` is vulnerable to self-enhancement bias. However, with the above debiasing prompt, in Table 7 shows that the results of new prompts are highly similar to the original prompts. Similarly, Table 6 shows that the results of modified and original prompts are differ only lightly.

We also designed a second methodology to mitigate self-enhancement bias. In this experiment, multiple LEA models evaluate the performance of IQA models during each interaction ("Multi-perspective"). In other words, we introduced third-party evaluations, where various LEA models assess the IQA models' performance instead of relying solely on the LEA model itself involved in the interaction. After evaluation, we use the average score from all LEA models as the final score. The results of IQA-Eval-Multi-perspective look as in Table 8. The correlations between IQA-Eval-Multi-perspective and human evaluations are in Table 9.

We believe that that the self-preference bias has limited impact on IQA-Eval.

Table 8: IQA-EVAL-Multi-Perspective Results of IQA Models. MP indicates "Multi-Perspective". Bold numbers indicate they are the closest to human results.

| LEA models | Helpfulness | | | Fluency | | | # Queries | | | Accuracy | | |
|---|---|---|---|---|---|---|---|---|---|---|---|---|
| | TDA | TB | DA | TDA | TB | DA | TDA | TB | DA | TDA | TB | DA |
| Human | 4.60 | 3.84 | 3.52 | 4.35 | 3.84 | 3.22 | 1.78 | 2.57 | 2.66 | 0.69 | 0.52 | 0.48 |
| IQA-EVAL-GPT4-MP | **4.32** | **3.70** | **3.53** | 4.57 | **3.74** | 3.68 | **1.57** | 2.27 | **2.37** | 0.87 | 0.83 | 0.67 |
| IQA-EVAL-Claude-MP | 3.96 | 3.13 | 3.10 | **4.29** | 3.51 | **3.22** | 2.20 | **2.67** | 2.07 | **0.67** | **0.53** | 0.57 |
| IQA-EVAL-GPT3.5-MP | 3.98 | 3.23 | 3.04 | **4.41** | 3.67 | 3.59 | **1.57** | 1.77 | 2.00 | 0.63 | 0.47 | **0.53** |

Table 9: Pearson Correlation ($\rho$) between IQA-EVAL-Multi-Persepctive evaluations and human judgments.

| LEA models | Helpfulness | Fluency | Overall |
|---|---|---|---|
| IQA-EVAL-GPT4-MP | **0.702** | 0.601 | **0.624** |
| IQA-EVAL-Claude-MP | 0.663 | **0.613** | 0.602 |
| IQA-EVAL-GPT3.5-MP | 0.641 | 0.552 | 0.533 |

## 6.5 Analysis

**Stronger IQA models require fewer turns in interactions.** On the more challenging AmbigQA, the stronger model, GPT4, typically requires only one turn to assist LEA in solving questions with high accuracy. In contrast, less capable models like Llama2 and GPT3.5 need more turns to clarify ambiguous entities and have lower QA accuracies. A similar trend is observed on the HotpotQA.

**We obtain a similar model ranking with a much lower cost.** Compared to Chatbot Arena [4], our accuracy-based ranking of IQA Models follows a similar trend: GPT4 > Claude > GPT3.5 > Llama2 > Zephyr. In addition, our evaluation method, IQA-EVAL, is fully automated. Our method makes it a cost-effective alternative for large-scale evaluations.

Table 10: Accuracy of IQA Models (recent LLMs) on two datasets (Non-interactive setting).

| IQA Models | HotpotQA | AmbigQA |
|---|---|---|
| GPT3.5 | 0.43 | 0.62 |
| GPT4 | **0.46** | **0.63** |
| Claude | 0.28 | 0.5 |
| Llama2 | 0.24 | 0.29 |
| Zephyr | 0.25 | 0.31 |

**Evaluation of interaction performance does not always match Non-Interaction performance.** In interaction evaluations, accuracy on final results is not the only metric to show IQA Models' performance. The quality of intermediate responses is a significant aspect. On both "helpfulness" and "fluency" metrics, Claude is always the best IQA Model on HotpotQA questions, while on AmbigQA, Llama2 and GPT3.5 outperform GPT4. IQA Model rankings on these two aspects differ from those in Chatbot Arena (non-interaction).

**The performance of IQA Models largely affects the final performance.** The accuracies in both Table 5 and Table 10 show a consistent trend. The Pearson correlations of the accuracy between the tables are 0.77 and 0.87 on both datasets, respectively. A strong IQA model, such as GPT4, can lead the LEA to finish tasks and largely improve the LEA's performance on those tasks. Weak assistants may drag down the LEA's performance, such as the performance of the LEA on both datasets decreases after interacting with Zephyr.

## 7 Conclusion

To conclude, we introduced IQA-EVAL, a novel approach for evaluating interactive question-answering systems using large language models. Our methodology achieves automatic interaction generation and evaluation with LEA, and enhances the evaluation process by assigning personas to LEA for better matching diverse groups of people. We show that our approach aligns closely with real human interactions and judgment, indicating that a scalable, automatic IQA-EVAL process can be achieved. We providing insights on recent LLM's capability in conducting IQA with IQA-EVAL which would cost $5,000 for human evaluations.

---

[4] https://huggingface.co/spaces/lmsys/chatbot-arena-leaderboard

## Acknowledgement

We thank the anonymous reviewers for valuable and insightful feedback. This research is supported in part by the National Science Foundation CAREER Grant IIS-2340435, Amazon Research Award and Cisco Research Award. Any opinions, findings, and conclusions or recommendations expressed herein are those of the authors and do not necessarily represent the views, either expressed or implied, of the U.S. Government.

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

# A Limitations

In conducting this study, certain limitations have influenced our scope and findings. First and foremost, LLMs are shown to demonstrate self-favouring behaviours [Panickssery et al., 2024], and no verified or accessible mitigations to this issue exist to the best of our knowledge. We discuss some of our attempts to address this in Section 6.4, but this limitation necessitates future research.

Our methodology was applied exclusively to multi-choice question-answering tasks due to constraints imposed by the datasets used. Moreover, we do not investigate how allowing prompt editing could affect the results. This choice limits the generalizability of our findings across the wider array of question answering formats that exist in both academic research and practical applications.

We advocate for subsequent research efforts to extend the application of our proposed evaluation strategies to a more diverse set of question answering tasks, beyond the multi-choice format.

Furthermore, there is a significant opportunity to test these methods with a broader spectrum of LLMs, including those at the cutting edge of the field. Such expansions would not only validate the versatility and robustness of our approaches but also potentially uncover additional insights into the nuances of LLM interaction and performance in varied contexts.

# B Ethics Statement

In our study, we meticulously crafted each persona to mitigate bias, ensuring they do not adversely impact the IQA-Eval process, as evidenced by the results presented in the table above. However, our personas represent only a limited range. There exists the potential for negative effects from other personas that may be inadequately designed or deliberately biased to achieve specific outcomes. Thus, we oppose irresponsible persona designs that result in biased evaluation results. The principle of persona definition and design should be thoroughly studied in future works. We hope our work will help facilitate future research into better automatic interaction evaluations aligning with crowds.

# C Prompts

Both C.1 and C.2 are adapted from Lee et al. [2023]. All the following prompts are from the view of LEA in interactions.

## C.1 Interaction Evaluation Prompt

We follow the setting in the data and evaluate conversations worker-wise. In other words, we combine conversations for each worker and send all of them to evaluators. The prompt we send to APIs follows

the following format:

```
You are a helpful and precise assistant for checking the quality
of the AI assistant's responses in interactions.

{Question 1}
{Golden Answer 1}
{Conversation 1}
{User Answer 1}
...
{Question n}
{Golden Answer n}
{Conversation n}
{User Answer n}

Please evaluate the above conversations between user and AI assistant
by using the following metrics:
Fluency (5-point Likert):  How clear (or fluent) were the responses
from the AI Assistant?
Helpfulness (5-point Likert):  Independent of its fluency, how helpful
was having access to the AI Assistant compared to not having access?
Helpfulness (free-form):  Why did you find the AI Assistant helpful
or unhelpful?
Please output each of the above metrics line-by-line.
```

## C.2 Interaction Generation Prompt

Since this is a multi-choice question answering task, the full prompt for **models** is as follows:

```
You are mimicking a human.
You are trying to choose the correct answer to the given question.
Please ask an assistant sub-questions for help approaching answers.
In each turn, please only ask one sub-question to interact with an
assistant.  In the sub-questions, please include all necessary
information, such as the question and options, in the original
question.  If you know the answer, please output "So, the answer
is:  A, B, C, or D."

{QA Question and choices}
{User Model's query:  [question 1]}
{Assistant's answer:  [answer 1]}
{User Model's query:  [question 2]}
{Assistant's answer:  [answer 2]}
...
{User Model's query:  [question n]}
{Assistant's answer:  [answer n]}
{User Model's final answer}
```

If the current turn reaches the maximum number we set, the system prompt before "{Question}" looks as follows:

```
Please choose the correct answer to the given question.  Please
output "So, the answer is:  A, B, C, or D."
```

### C.2.1  QA Question and choices format

The question prompt for multi-choice questions in MMLU is as follows:

```
<question>
A. <option A>
B. <option B>
C. <option C>
D. <option D>
```

For HotpotQA and AmbigQA datasets, the question prompt only contains a question, such as:
```
<question>
```

### C.3  Persona Prompts

We design distinct prompts for each persona. Both prompts in meta-evaluation and model interaction modules change with personas.

In model interaction prompts, we only modify the first sentence based on personas. See all persona prompts in Table 11.

Table 11: Persona Interaction and Evaluation Descriptions

| Persona | Persona Interaction Description | Persona Evaluation Description |
|---|---|---|
| Expert | You are mimicking a knowledgeable human who can quickly understand new concepts. You need help from an assistant to learn and answer questions. | The AI Assistant helps a knowledgeable human to answer a question. The assistant should provide straightforward, informative, and in-depth answers to human questions. |
| Critical-Thinker | You are mimicking a human who prefers interactions rich in critical information. You need help from an assistant and try to get critical information from it to answer the following questions. | The AI Assistant should provide clear, non-vague, and precise information or options and help user deduce answers. (Detailed evaluation criteria were indicated but not fully transcribed due to length.) |
| Adaptability-Seeker | You are mimicking a human who prefers an adaptable assistant who can always understand his questions. You need help from an assistant to answer questions. | The AI Assistant helps a human who prefers an adaptable assistant. The assistant should understand user's questions, provide related options, and help user deduce answers. |
| Clarity-Seeker | You are mimicking a human who prefers clear information in conversations. You need help from an assistant and want to get clear information from it to answer questions. | The AI Assistant helps a human who prefers clear information in conversations. The AI should provide non-vague, precise information to help user deduce answers. |

# D   Additional Experiments on Natural Questions

We benchmark IQA models in another dataset called Natural Questions (Kwiatkowski et al. [2019b]). This dataset comprises authentic questions posed by users about Wikipedia articles, demanding true multi-turn dialogues for resolution, akin to the setup in multi-turn conversational QA dataset QuAC (Choi et al. [2018]). The experiment results are as in Tables 12 and 13. All numbers of queries in the two tables are around 3, and each response to a query from IQA models contains an average of 2 sentences.

Given the number of sentences in each IQA model's response in Table 14, the non-interactive outputs are roughly equivalent to about two interaction turns, less than three turns in interactive outputs. Thus, The interaction process of IQA-Eval involves not only reasoning processes but also simulating genuine interactive multi-turn conversations. This suggests that the performances shown in tables 12 and 13 above are driven more by multi-turn interactions than by reasoning processes. Furthermore, these interactions lead to enhanced accuracy, as demonstrated by the superior results in the first two tables compared to those in the last table (non-interactive).

# E   Sensitive to Persona Distributions

We conduct two new experiments to study the effects of changing persona assignments. Results indicate that IQA-Eval is sensitive to incorrect persona assignments. When the persona distribution

Table 12: IQA-Eval benchmarking results on the Natural Questions by Claude-3

| IQA models | Helpfulness | Fluency | # Queries | Accuracy |
|---|---|---|---|---|
| GPT3.5 | **4.86** | **4.88** | **2.82** | 0.42 |
| Claude | 4.88 | 4.90 | 3.02 | 0.38 |
| Llama2 | 4.90 | 4.84 | 3.18 | 0.34 |
| Zephyr | 4.84 | 4.90 | 3.02 | 0.28 |

Table 13: IQA-Eval benchmarking results on the Natural Question by GPT-4

| IQA models | Helpfulness | Fluency | # Queries | Accuracy |
|---|---|---|---|---|
| GPT3.5 | **4.12** | **5.00** | **2.76** | 0.44 |
| Claude | 4.02 | 5.00 | 2.76 | 0.40 |
| Llama2 | 3.20 | 4.84 | 3.08 | 0.32 |
| Zephyr | 3.30 | 4.86 | 2.92 | 0.36 |

Table 14: Average number of sentences and accuracy scores of IQA Models (non-interactive setting)

| IQA models | # Sentences | Accuracy |
|---|---|---|
| GPT3.5 | 4.66 | 0.38 |
| Claude | 3.16 | 0.34 |
| Llama2 | 5.21 | 0.30 |
| Zephyr | 4.68 | 0.24 |

Table 15: IQA-EVAL results under different persona distribution on the expert persona.

| LEA models | Helpfulness | | | | Fluency | | | |
|---|---|---|---|---|---|---|---|---|
| | TDA | TB | DA | $\rho$ | TDA | TB | DA | $\rho$ |
| Human | 4.60 | 3.84 | 3.52 | | 4.35 | 3.84 | 3.22 | |
| IQA-EVAL (Expert) | 4.17 | 3.08 | 3.12 | 0.756 | 4.47 | 3.84 | 3.40 | 0.787 |
| IQA-EVAL (20% Expert) | 4.31 | 3.26 | 3.44 | 0.708 | 4.62 | 4.09 | 3.65 | 0.741 |
| IQA-EVAL (40% Expert) | 4.21 | 3.14 | 3.23 | 0.751 | 4.49 | 3.88 | 3.44 | 0.779 |
| IQA-EVAL (60% Expert) | 4.11 | 3.01 | 3.00 | 0.725 | 4.43 | 3.77 | 3.34 | 0.734 |
| IQA-EVAL (80% Expert) | 4.02 | 2.90 | 2.79 | 0.680 | 4.30 | 3.56 | 3.12 | 0.703 |
| Human (Pure Expert) | 4.69 | 4.00 | 3.73 | | 4.36 | 3.96 | 3.26 | |
| IQA-EVAL (Pure Expert) | 4.37 | 3.57 | 3.33 | 0.778 | 4.20 | 3.40 | 2.97 | 0.786 |

is incorrect (such as 20% Expert in Table 15), the performance of IQA-EVAL shows a lower correlation with human evaluations.

Moreover, the last two lines in Table 15 describe the correlation between human evaluations and IQA-EVAL within a sub-group only containing pure experts. The correlation results in line "IQA-EVAL (Pure Expert)" represent that (1) our personas accurately represent the pure expert group, as its correlation with the line "Human (Pure Expert)" remains nearly consistent with those in line "IQA-EVAL (Expert)" and (2) given this completely correct persona distribution, our IQA-EVAL correlates well with human evaluations.

## F  Accurate QA models are preferred by humans in IQA-EVAL

The quote from our cited paper Lee et al. [2023] is "[...] perception of helpfulness is not necessarily reflected in the overall interaction accuracy." It describes the conclusion of multiple tasks in that paper (e.g. text summarization, social dialogue, QA). However, in the QA settings, Table 3 in Lee et al. [2023] shows that humans prefer accurate models on the QA task.

Table 16: Evaluation results of interactions between LEA and IQA models.

| LEA models | Helpfulness | | | | Fluency | | | | Accuracy | | | |
| --- | --- | --- | --- | --- | --- | --- | --- | --- | --- | --- | --- | --- |
| | GPT 3.5 | Claude -instant | Llama2 -8b | Zephyr -Alpha | GPT 3.5 | Claude -instant | Llama2 -8b | Zephyr -Alpha | GPT 3.5 | Claude -instant | Llama2 -8b | Zephyr -Alpha |
| IQA-EVAL-GPT4 | 4.60 | 4.60 | 3.83 | 4.27 | 4.97 | 5.00 | 4.87 | 4.93 | 0.93 | 0.93 | 0.83 | 0.93 |
| IQA-EVAL-Claude | 4.90 | 5.00 | 4.97 | 4.97 | 4.87 | 5.00 | 4.93 | 4.87 | 0.73 | 0.8 | 0.57 | 0.73 |

Table 17: Evaluation results of non-interactions (direct answers) between LEA and IQA models.

| LEA models | Helpfulness | | | | Fluency | | | | Accuracy | | | |
| --- | --- | --- | --- | --- | --- | --- | --- | --- | --- | --- | --- | --- |
| | GPT 3.5 | Claude -instant | Llama2 -8b | Zephyr -Alpha | GPT 3.5 | Claude -instant | Llama2 -8b | Zephyr -Alpha | GPT 3.5 | Claude -instant | Llama2 -8b | Zephyr -Alpha |
| IQA-EVAL-GPT4 | 4.33 | 4.17 | 2.70 | 3.53 | 5.00 | 4.97 | 4.13 | 4.33 | 0.83 | 0.80 | 0.47 | 0.57 |
| IQA-EVAL-Claude | 4.97 | 5.00 | 4.53 | 4.87 | 4.97 | 4.97 | 4.47 | 4.97 | 0.83 | 0.80 | 0.47 | 0.57 |

We also conducted experiments (1) using LEA to evaluate interactions between LEAs and IQA models (interactive) and (2) using LEA to evaluate direct answers generated by IQA models (non-interactive). Our experiments in Tables 16 and 17 show that LEA models prefer accurate models, which aligns well with the conclusion from human annotations.

# G    Bias Evaluation

We follow the method proposed by Sheng et al. [2021] and conduct a new experiment to evaluate the offensiveness and harmfulness of our personas using the RealToxicityPrompts dataset (Gehman et al. [2020]) on our LEA models. The results are in the table 18. The values in the table above represent the success rates (higher is better) for each bias metric, persona, and LEA model (`GPT3.5` and `Claude`). Scores labeled "None" are consistently lower than those for all personas, indicating that our personas do not increase offensiveness or harmfulness in conversations.

Table 18: Evaluating persona biases on offensiveness and harmful metrics. A high score indicates better results.

| Persona | Offensiveness | | Harmful | |
| --- | --- | --- | --- | --- |
| | IQA-EVAL-GPT3.5 | IQA-EVAL-Claude | IQA-EVAL-GPT3.5 | IQA-EVAL-Claude |
| None | 89.5 | 91.7 | 62.5 | 72.5 |
| Expert | 95.5 | 97.3 | 67.8 | 75.5 |
| Critical-Thinker | 93.3 | 95.5 | 65.4 | 73.7 |
| Adaptability-Seeker | 94.5 | 95.5 | 62.8 | 74.6 |
| Clarity-Seeker | 95.0 | 94.0 | 62.5 | 73.0 |

