# OpenReview forum: "IQA-EVAL: Automatic Evaluation of Human-Model Interactive Question Answering"
_NeurIPS.cc/2024/Conference — NeurIPS 2024 poster_

### Official Review · Reviewer_RXxM · 2024-07-07

**Soundness:** 2
**Presentation:** 3
**Contribution:** 2
**Rating:** 5
**Confidence:** 3

**Summary:**

This paper proposes an automated evaluation framework for Interactive Question Answering tasks based on LLM-based agent. The authors utilize LLMs to simulate the interaction between human and IQA models and use them to evaluate the interactions automatically. Additionally, they assign predefined personas to LLMs to better simulate the interaction characteristics of different groups. Their evaluation framework achieves a stronger correlation with human evaluations and eliminates the high costs of manual evaluation.

**Strengths:**

The strength of this paper is that it highlighted the importance of interaction in evaluation process for QA tasks and provided a solution for this problem, showing new insights into the evaluation for QA tasks.

**Weaknesses:**

The weakness of this paper is the lack of technical contributions and novelty, and its experimental analysis is not strong enough. They only used prompts to implement their methods, without any prompt engineering approach. Furthermore, their idea of using LLMs as evaluators is similar to some previous research papers, such as G-Eval [1].

[1] Y. Liu, D. Iter, Y. Xu, S. Wang, R. Xu, and C. Zhu. G-eval: Nlg evaluation using gpt-4 with better human alignment. CoRR, abs/2303.16634, 2023.

**Questions:**

Major questions:
1. The paper utilizes LLMs as evaluators for Interactive Question Answering. However, according to some previous research papers [2], a systematic bias may be introduced by LLM evaluators and the results of evaluation may be influenced. Did you consider the impact of bias on the results in your research and try any methods to eliminate it?
2. You mentioned that accurate models are not necessarily preferred by humans in your paper, but you did not demonstrate this result in your experiments. Could you add a control group in your experiment?
3. There exists some errors in your analysis of LEA for stage 2. You stated that models’ evaluations highly correlate with human evaluation, while the Pearson correlation coefficient between 0.2 and 0.5 only indicates a weak or moderate correlation.
4. In your analysis of Table 1, you concluded that model gave higher scores for the “Fluency” metric than human because of the clarity and grammatical correctness of the response generated by IQA models. Could you please provide a detailed explanation of why you concluded that the model understands the concept of “Fluency” better than humans?

Minor questions:
1. Could you please point out the LEA model used in the experiments shown in Table 3, like in Table 1 and 2?

[2] P. Wang, L. Li, L. Chen, D. Zhu, B. Lin, Y. Cao, Q. Liu, T. Liu, and Z. Sui. Large language models are not fair evaluators. CoRR, abs/2305.17926, 2023.

**Limitations:**

Yes

---

> ### Author Rebuttal · Authors · 2024-08-07
>
> Thank you Reviewer RXxM for your reviews. Below we'd like to address your concerns.
> # Weaknesses
>
> ----
> > **This paper's weakness is its lack of technical contributions and novelty, and its experimental analysis is not strong enough. It only used prompts to implement its methods without any prompt engineering approach.**
>
> Please refer to the generel rebuttal above.
>
> ----
> > **Their idea of using LLMs as evaluators is similar to some previous research papers, such as G-Eval**
>
> G-Eval implements automatic evaluation for direct outputs from Large Language Models (LLMs) which is non-interactive. However, chat interactions involving multiple turns are more prevalent in real-world applications. Currently, no method is available to automatically evaluate the performance of models in such interactive settings [3]. Our approach, for the first time, addresses this gap by using LLMs to automatically assess the performance of assistant models during interactive scenarios. Moreover, we incorporate personas in our IQA-Eval to better represent the crowd, further significantly improving correlations.
>
> # Questions
>
> ----
> > **A systematic bias may be introduced by LLM evaluators and the results of evaluation may be influenced. Did you consider the impact of bias on the results in your research and try any methods to eliminate it?**
>
> We acknowledge this is a common conern. Please refer to the "Concerns on self-favoring bias" section in the general rebuttal.
>
> ----
> > **You mentioned that accurate models are not necessarily preferred by humans in your paper, but you did not demonstrate this result in your experiments. Could you add a control group in your experiment?**
>
> Thank you for pointing out this. We are sorry for this inaccurate statement. We will add this accurate claim in the paper: accurate QA models are preferred by humans in interaction-aware evaluations.
>
> The quote from our cited paper [3] is “[...] perception of helpfulness is not necessarily reflected in the overall interaction accuracy.” It describes the conclusion of multiple tasks in that paper (e.g. text summarization, social dialogue, QA). However, in the QA settings, Table 3 in Lee et al. (2023) shows that humans prefer accurate models on the QA task.
>
> We also conducted **new experiments** **(1)** using LEA to evaluate interactions between LEAs and IQA models (interactive) and **(2)** using LEA to evaluate direct answers generated by IQA models (non-interactive). Our experiments show that **LEA models prefer accurate models, which aligns well with the conclusion from human annotations.**
>
> (1) Evaluate interactions between LEA and IQA models.
>
> ||||||||||||||
> |---|---|---|---|---|---|---|---|---|---|---|---|---|
> |5-point Likert|Helpfulness||||Fluency||||Accuracy||||
> |LEA models|GPT-3.5|Claude-instant|Llama2-8b|Zephyr-Alpha|GPT-3.5|Claude-instant|Llama2-8b|Zephyr-Alpha|GPT-3.5|Claude-instant|Llama2-8b|Zephyr-Alpha|
> |IQA-EVAL-GPT4|4.60|4.60|3.83|4.27|4.97|5.00|4.87|4.93|0.93|0.93|0.83|0.93|
> |IQA-EVAL-Claude|4.90|5.00|4.97|4.97|4.87|5.00|4.93|4.87|0.73|0.8|0.57|0.73|
>
> (2) Evaluate direct answers generated by IQA models
> ||||||||||||||
> |---|---|---|---|---|---|---|---|---|---|---|---|---|
> |5-point Likert|Helpfulness||||Fluency||||Accuracy||||
> |LEA models|GPT-3.5|Claude-instant|Llama2-8b|Zephyr-Alpha|GPT-3.5|Claude-instant|Llama2-8b|Zephyr-Alpha|GPT-3.5|Claude-instant|Llama2-8b|Zephyr-Alpha|
> |IQA-EVAL-GPT4|4.33|4.17|2.70|3.53|5.00|4.97|4.13|4.33|0.83|0.80|0.47|0.57|
> |IQA-EVAL-Claude|4.97|5.00|4.53|4.87|4.97|4.97|4.47|4.97|0.83|0.80|0.47|0.57|
>
> ----
> ----
> > **There exists some errors in your analysis of LEA for stage 2. You stated that models’ evaluations highly correlate with human evaluation, while the Pearson correlation coefficient between 0.2 and 0.5 only indicates a weak or moderate correlation**
>
> Thank you for bringing this to our attention. We will modify the claimed relation to be weak to moderate in the paper.
>
> For interactions between humans and IQA models (Figure 2), LEA evaluations moderately correlate with human evaluations. However, for interactions between LEAs and IQA models (Table 2) which is the main focus of our paper, LEA evaluations highly correlate (around 0.6) with human evaluations. This indicates that **LEA models are helpful when participating in the whole evaluation process**, including both the interaction and evaluation. However, **when evaluating interactions between human and IQA models, LEAs focus differently from humans**, which causes moderate correlations in the “analysis for LEA for stage 2”.
>
> ----
> > Unsupported paper claim: **the model understands the concept of “Fluency” better than humans?**
>
> Sorry for the confusion. We did not conclude that the model understands the concept of “Fluency” better than humans. Instead, we wanted to emphasize that **IQA-Eval scores on “Fluency” are close and highly correlated to human judgments**. Both scores given by human and LEA models show that IQA models provide fluent outputs. We will clarify and update this claim in the final version.
>
> ----
> > **Could you please point out the LEA model used in the experiments shown in Table 3, like in Table 1 and 2?**
>
> In Table 3, the LEA model is GPT-3.5-turbo-1106. We will update it in the final version.
>
> ----
> **References:**
> 1. Zheng, Lianmin, et al. "Judging llm-as-a-judge with mt-bench and chatbot arena." Advances in Neural Information Processing Systems 36 (2024).
>
> 2. Furniturewala, Shaz, et al. "Thinking Fair and Slow: On the Efficacy of Structured Prompts for Debiasing Language Models." arXiv preprint arXiv:2405.10431 (2024).
>
> 3. Lee, Mina, et al. "Evaluating human-language model interaction." arXiv preprint arXiv:2212.09746 (2022).
>
> 4. Liu, Yang, et al. "G-eval: Nlg evaluation using gpt-4 with better human alignment." arXiv preprint arXiv:2303.16634 (2023).

---

> > ### Author Response · Authors · 2024-08-10
> > **Shall we have more discussions?**
> >
> > Dear Reviewer,
> >
> > Do you have any other questions of interest? In our rebuttal, we have added relevant explanations and several experiments, and we believe that these can to some extent address your concerns.
> >
> > We are eagerly looking forward to your response!

---

> > > ### Author Response · Authors · 2024-08-12
> > > **More discussions**
> > >
> > > Dear Reviewer,
> > >
> > > The discussion period is ending.
> > >
> > > Do you have any other questions of interest? In our previous rebuttal, we have added relevant explanations and several experiments, and we believe that these can to some extent address your concerns.
> > >
> > > We are eagerly looking forward to your response!

---

> > > > ### Comment · Area_Chair_oCjR · 2024-08-12
> > > > **Reviewer please respond**
> > > >
> > > > Dear reviewer,
> > > >
> > > > Thank you for your efforts in reviewing this paper. Now that the authors have provided their response, do you have any further comments?
> > > >
> > > > Thank you,
> > > > AC

---

> > > > ### Author Response · Authors · 2024-08-13
> > > > **More discussions?**
> > > >
> > > > Dear Reviewer,
> > > >
> > > > The discussion period is ending.
> > > >
> > > > Do you have any other questions of interest? In our previous rebuttal, we have added relevant explanations and several experiments, and we believe that these can address your concerns.
> > > >
> > > > We are eagerly looking forward to your response!

---

> > > > > ### Comment · Reviewer_RXxM · 2024-08-14
> > > > > **Response to the authors**
> > > > >
> > > > > Sorry for the late reply! Thank you for your reply, my concern has been addressed. I have raised my score.

---

### Official Review · Reviewer_unmC · 2024-07-11

**Soundness:** 3
**Presentation:** 3
**Contribution:** 2
**Rating:** 5
**Confidence:** 4

**Summary:**

This paper introduces IQA-EVAL, a framework for automatically evaluating interactive question-answering (IQA) systems using large language models. The authors propose using LLM-based Evaluation Agents (LEAs) to simulate human behavior in both generating interactions with IQA models and evaluating those interactions. The framework also incorporates persona assignments to LEAs to better represent diverse user groups. The authors demonstrate that IQA-EVAL achieves high correlation with human judgments and use it to benchmark several recent LLMs on complex question-answering tasks.

**Strengths:**

- Proposes a fully automated framework for evaluating interactive QA systems, addressing the need for more efficient evaluation methods
- Incorporates persona assignments to LEAs, allowing for more nuanced and diverse simulations of user interactions
- Demonstrates strong correlation with human judgments

**Weaknesses:**

- Limited novelty, as using LLMs as judges is quite common in evaluation tasks. This paper mainly focuses on a relatively new setting - the interactive QA setting.
- The considered interaction scenarios could be more diverse to better reflect real-world user behaviors

**Questions:**

- Line 207~213: can you provide more explanations into why ChatGPT sometimes outperformed GPT-4 in your experiments? This seems counterintuitive given GPT-4's generally stronger capabilities.
- Have you considered incorporating more diverse interaction patterns, such as clarification questions, ambiguous queries, or follow-up questions? How might this affect the evaluation results?
- Have you explored how different persona distributions might impact the evaluation results? How sensitive is the framework to changes in persona assignments?

**Limitations:**

The authors have made some effort to address limitations, such as acknowledging potential biases in LLM-based evaluation and discussing the impact of persona assignments. However, the paper would benefit from discussing the limited diversity of interaction patterns considered.

---

> ### Author Rebuttal · Authors · 2024-08-07
>
> Thank you for your helpful review. We would like to address the mentioned weakness and questions below.
>
> ----
> # Weaknesses:
>
> > **Limited novelty, as using LLMs as judges is quite common in evaluation tasks. This paper mainly focuses on a relatively new setting - the interactive QA setting.**
>
> Previous works, like G-Eval, implement automatic evaluation for direct outputs from Large Language Models (LLMs) which is non-interactive. However, chat interactions involving **multiple turns are more prevalent in real-world applications**, as recognized by the HCI community [1]. Currently, **no method is available to automatically evaluate the performance of models in such interactive settings**. Our approach, for the first time, addresses this gap by using LLMs to automatically assess the performance of assistant models during interactive scenarios.
>
> Additionally, **for the first time, we augment LLM-agent with personas** in our IQA-Eval to better represent the crowd, further significantly improving correlations.
>
> ----
> > **The considered interaction scenarios could be more diverse to better reflect real-world user behaviors**
>
> Please see the reply to question 2 Below.
>
>
> ----
> # Questions:
>
> > **Line 207~213: can you provide more explanations into why ChatGPT sometimes outperformed GPT-4 in your experiments? This seems counterintuitive given GPT-4's generally stronger capabilities.**
>
>
> Table 1 evaluates how close evaluators are to humans on absolute scores. Based on the results, although GPT-4 outperforms GPT-3.5 in accuracy and demonstrates stronger capabilities, scores given by GPT-3.5 are closer to humans than GPT-4. The main reason for low GPT-4 scores is its **more critical and strict evaluation** of “Helpfulness”.
>
> Although giving low scores, Table 2 shows that **GPT-4 mostly correlates to human evaluations** since its scores have the most similar trend as human scores. Given GPT-4’s generally strong capability, this result is not counterintuitive and supports the above reason.
>
>
> ----
> > **Have you considered incorporating more diverse interaction patterns, such as clarification questions, ambiguous queries, or follow-up questions? How might this affect the evaluation results?**
>
>
> In section 5, we discussed the effect of different types of persona and more diverse interaction patterns (i.e. clarification, ambiguous, complex, follow-up, expert, detailed, straightforward, knowledgable, etc.). For example, each persona produces different interaction patterns which are diverse. The persona “Clarity-Seeker” prefers to clarify questions, terminologies, proper nouns, etc. The persona “Adaptability-Seeker” always proposes ambiguous questions and prefers IQA models that can understand its questions. The persona “Critical-Seeker” always follows the original question and asks critical questions to IQA models to answer the question. Our different personas have covered a variety of interaction patterns. Results are shown in Tables 3 and 4. **By adding personas and having more diverse interactions, IQA-Eval achieves higher correlations with human evaluations and better represents the crowd**.
>
> Moreover, in section 6.2 (LLM to Benchmarks), we benchmark IQA models using our IQA-EVAL **on question answering data and ambiguous queries**, showing our IQA-EVAL can be applied to different types of questions and interaction patterns and provide useful feedback.
>
>
> ----
> > **Have you explored how different persona distributions might impact the evaluation results? How sensitive is the framework to changes in persona assignments?**
>
>
> In practice, the distribution of personas should be thoroughly surveyed. Given those distributions, the evaluation performance of IQA-Eval should always be similar to human evaluations (high correlation scores). **We expect that the distribution of personas should have little effect on IQA-Eval since all scores well in correlation with human raters (0.634-0.690)**.
>
> ----
> ## References:
>
> 1. Lee, Mina, et al. "Evaluating human-language model interaction." arXiv preprint arXiv:2212.09746 (2022).

---

> > ### Author Response · Authors · 2024-08-07
> > **More clarification on the reply to question 3**
> >
> > We are sorry for the minor inaccuracy in the original reply to question 3. We expect that IQA-EVAL is sensitive to incorrect persona assignments.
> >
> > We conduct two new experiments to study the effects of changing persona assignments. When the persona distribution is incorrect (such as 20% Expert in the table below), the performance of IQA-EVAL shows a lower correlation with human evaluations.
> >
> > Moreover, the last two lines in the following table describe the correlation between human evaluations and IQA-EVAL within a sub-group only containing pure experts. The correlation results in line “IQA-EVAL (Pure Expert)” represent that (1) our personas accurately represent the pure expert group, as its correlation with the line “Human (Pure Expert)” remains nearly consistent with those in line “IQA-EVAL (Expert)” and (2) given this completely correct persona distribution, our IQA-EVAL correlates well with human evaluations.
> >
> > | 5-point Likert                | Helpfulness |      |      |       | Fluency |      |      |       |
> > | ----------------------------- | ----------- | ---- | ---- | ----- | ------- | ---- | ---- | ----- |
> > | LEA models                    | TDA         | TB   | DA   | ρ     | TDA     | TB   | DA   | ρ     |
> > | Human                         | 4.60        | 3.84 | 3.52 |       | 4.35    | 3.84 | 3.22 |       |
> > | IQA-EVAL (Expert)             | 4.17        | 3.08 | 3.12 | 0.756 | 4.47    | 3.84 | 3.40 | 0.787 |
> > | IQA-EVAL (20% <br><br>Expert) | 4.31        | 3.26 | 3.44 | 0.708 | 4.62    | 4.09 | 3.65 | 0.741 |
> > | IQA-EVAL (40% <br><br>Expert) | 4.21        | 3.14 | 3.23 | 0.751 | 4.49    | 3.88 | 3.44 | 0.779 |
> > | IQA-EVAL (60% <br><br>Expert) | 4.11        | 3.01 | 3.00 | 0.725 | 4.43    | 3.77 | 3.34 | 0.734 |
> > | IQA-EVAL (80% <br><br>Expert) | 4.02        | 2.90 | 2.79 | 0.680 | 4.30    | 3.56 | 3.12 | 0.703 |
> > | Human (Pure Expert)           | 4.69        | 4.00 | 3.73 |       | 4.36    | 3.96 | 3.26 |       |
> > | IQA-EVAL (Pure Expert)        | 4.37        | 3.57 | 3.33 | 0.778 | 4.20    | 3.40 | 2.97 | 0.786 |
> >
> > If you think we misunderstood your question, please let us know and feel free to ask any follow-up questions.

---

> > > ### Author Response · Authors · 2024-08-10
> > > **shall we have more discussions?**
> > >
> > > Dear Reviewer,
> > >
> > > do you have any other questions of interest? In our previous rebuttal, we have added relevant explanations and several experiments, and we believe that these can to some extent address your concerns.
> > >
> > > We are eagerly looking forward to your response!

---

> > > > ### Author Response · Authors · 2024-08-12
> > > > **More discussions**
> > > >
> > > > Dear Reviewer,
> > > >
> > > > The discussion period is ending.
> > > >
> > > > Do you have any other questions of interest? In our previous rebuttal, we have added relevant explanations and several experiments, and we believe that these can to some extent address your concerns.
> > > >
> > > > We are eagerly looking forward to your response!

---

> > > > > ### Comment · Area_Chair_oCjR · 2024-08-12
> > > > > **Reviewer please respond**
> > > > >
> > > > > Dear reviewer,
> > > > >
> > > > > Thank you for your efforts in reviewing this paper. Now that the authors have provided their response, do you have any further comments?
> > > > >
> > > > > Thank you,
> > > > > AC

---

> > > > > > ### Comment · Reviewer_unmC · 2024-08-13
> > > > > >
> > > > > > Thanks authors' for the detailed responses. While some of my questions have been addressed with more explanations, my overall opinions remain the same. I still think the contribution "augment LLM-agent with personas for eval" is a bit marginal, I tend to keep my score unchanged.

---

### Official Review · Reviewer_JewS · 2024-07-12

**Soundness:** 3
**Presentation:** 3
**Contribution:** 3
**Rating:** 7
**Confidence:** 3

**Summary:**

The authors introduce a novel method to simulate a human conversation when evaluated on an interactive question answering (IQA) and evaluate the simulated interaction according to some predefined metrics.

**Strengths:**

- The presentation of the paper is clear and easy to follow
- The paper is well-motivated. It recognizes how costly and time-consuming it is to use human evaluations
- The idea is reasonable

**Weaknesses:**

When using LLMs to evaluate the outputs from LLMs, recent research has shown LLMs to be biased in preferring their own generations compared to generations from other models, even if their generation wasn’t better [0]. This may bias the IQA-EVAL metrics if the same model is used for both IQA-EVAL and the IQA model.

The authors provided limited evidence for their claims in section 4.3. Could the authors provide a more robust justification of why “ChatGPT’s assessments align well with human evaluations”, and other claims made in this section?

The authors did not recognise that this method could cause negative societal impacts, instead stating “Evaluation works bear little risks for negative scoeital impacts”. LLMs have been repeatedly shown to exhibit strong biases as a result of their training procedure. Using LLMs for evaluation opens the risk of reinforcing these biases, whether within training of future models, or benchmark answers, or whatever is downstream of this method.

[0] https://arxiv.org/abs/2404.13076

**Questions:**

See Weaknesses

**Limitations:**

The authors addressed the limitations.

---

> ### Author Rebuttal · Authors · 2024-08-07
>
> Thank you for your helpful review. We would like to address the mentioned weakness and questions below.
>
>
> ----
>
> > **When using LLMs to evaluate the outputs from LLMs, recent research has shown LLMs to be biased in preferring their own generations compared to generations from other models, even if their generation wasn’t better. This may bias the IQA-EVAL metrics if the same model is used for both IQA-EVAL and the IQA model.**
>
> Please refer to the answer in the general author response on the top of the page.
>
>
>
> ----
> > **The authors provided limited evidence for their claims in section 4.3. Could the authors provide a more robust justification of why “ChatGPT’s assessments align well with human evaluations”, and other claims made in this section?**
>
> The content in section 4.3 contains our findings after reading LEA free-form outputs about “helpfulness”. The ChatGPT in section 4.3 only indicates GPT-3.5.
>
> For the claim ``ChatGPT’s assessments align well with human evaluations’’. In Table 1, GPT-3.5 always gives high sores, indicating it is more positive generally. Its scores are the closest to human scores in Table 1, and its correlation is also high in Table 2. Thus, we claim GPT-3.5 assessments align well with human evaluations.
>
>
> For the other claim, compared to GPT-3.5, Claude assigns lower scores. These results align with the claim that Claude flags more issues.
>
>
> ----
> > **The authors did not recognise that this method could cause negative societal impacts, instead stating “Evaluation works bear little risks for negative societal impacts”. LLMs have been repeatedly shown to exhibit strong biases as a result of their training procedure. Using LLMs for evaluation opens the risk of reinforcing these biases, whether within training of future models, or benchmark answers, or whatever is downstream of this method.**
>
>
>
> We apologize for the previous statement that “Evaluation works bear little risks for negative societal impacts”; this claim is inaccurate. Large Language Models (LLMs) have been repeatedly shown to exhibit strong biases due to their training procedures, which can result in significant societal impacts. These biases can perpetuate stereotypes, disadvantage certain groups, and influence downstream applications in harmful ways.
>
>
> To partially  address these concerns, we have implemented several measures to reduce bias in our evaluation process. For instance, we have fine-tuned prompts to minimize self-enhancement bias. Our experiments on the MMLU dataset demonstrate that bias does not significantly affect our evaluation results. Specifically, LEAs mainly evaluate the performance of IQA models in conversations. In IQA-Eval, each LEA model evaluates all IQA models, and since LEA models and IQA models are distinct, the impact of bias, such as LLMs favoring their own outputs, is mitigated in the evaluation results presented in sections 4 and 5.
>
>
>
> ----
> ## References:
>
> 1. Panickssery, Arjun, Samuel R. Bowman, and Shi Feng. "Llm evaluators recognize and favor their own generations." arXiv preprint arXiv:2404.13076 (2024).
>
> 2. Zheng, Lianmin, et al. "Judging llm-as-a-judge with mt-bench and chatbot arena." Advances in Neural Information Processing Systems 36 (2024).
>
> 3. Furniturewala, Shaz, et al. "Thinking Fair and Slow: On the Efficacy of Structured Prompts for Debiasing Language Models." arXiv preprint arXiv:2405.10431 (2024).

---

> > ### Comment · Reviewer_JewS · 2024-08-12
> >
> > I am grateful for the further analysis presented by the authors. Given their sound arguments, I am going to raise my score to a 7.

---

### Official Review · Reviewer_RAi2 · 2024-07-13

**Soundness:** 3
**Presentation:** 1
**Contribution:** 2
**Rating:** 6
**Confidence:** 4

**Summary:**

This research addresses the evaluation methodology for multi-turn conversation using Large Language Models (LLMs), a topic of active research recently. The study proposes an evaluation framework called IQA-Eval, which consists of the target model (IQA model) and an agent (LEA model) that engages in conversation with the IQA model and evaluates each turn. The primary challenge in multi-turn evaluation is to 1) interactively converse with the IQA model and 2) evaluate each generated turn in a cost and time-efficient manner. This research aims to overcome this challenge by leveraging LLMs that consider personas to facilitate both conversation and evaluation.
Through various experiments, the study confirms that applying LLMs for conversation and evaluation results in a high correlation with Human Evaluation. Additionally, it includes an analysis of the differences in the number of turns due to the varying capabilities of each IQA model.

**Strengths:**

1.Compared to existing LLM multi-turn evaluations, this approach enables faster and more economical assessments.
2.Examining the human correlation suggests that the evaluation is reliable. It is very promising in that it allows for automatic evaluation of interactive multi-turn conversation capabilities that resemble real-world scenarios.
3.The evaluation framework is very simple, making it adaptable for assessing various multi-turn situations.

**Weaknesses:**

- The evaluation dataset consists entirely of multiple-choice questions, making it unsuitable for generating and evaluating multi-turn conversations. This is evident in Tables 3 and 5, where the average length of conversation turns is less than 2. Such a situation suggests that the process might be more about reasoning rather than simulating genuine interactive multi-turn conversations. It is necessary to analyze whether the performance shown in Table 6 is due to multi-turn interactions or the influence of the reasoning path. Alternatively, an evaluation using datasets that assume genuine multi-turn dialogue scenarios is needed.
- The verification process for personas is lacking. There is no substantial evidence to consider that the performance differences by persona shown in Table 4 are reflective of the actual characteristics of those personas.
- A self-preference issue appears to have arisen in Table 5, and there is no control in place to address this.

**Questions:**

- How does the Free Form Feedback in this study differ from the explanations provided by existing LLM-based evaluation methods (such as LLM-Eval and G-Eval)?
- What is the total number of repetitions for the experiments mentioned in section 5.2? Could you provide the variance values for each experiment?
- Why are the results for AmbigQA not included? Even if the performance of the IQA models is not optimal, the results should still be presented.

**Limitations:**

- The limitations are stated in the Appendix, clearly outlining the limitations of the experiments.

---

> ### Author Rebuttal · Authors · 2024-08-07
>
> Thank you for your insightful review. We would like to address your concerns as follows:
>
> # Weaknesses:
>
> >1. The evaluation dataset consists entirely of multiple-choice questions, making it unsuitable for generating and evaluating multi-turn conversations. This is evident in Tables 3 and 5, where the average length of conversation turns is less than 2. Such a situation suggests that the process might be more about reasoning rather than simulating genuine interactive multi-turn conversations. It is necessary to analyze whether the performance shown in Table 6 is due to multi-turn interactions or the influence of the reasoning path. Alternatively, an evaluation using datasets that assume genuine multi-turn dialogue scenarios is needed.
>
> We focus on task-driven dialogues. In the IQA setting, humans interact with assistant models to answer questions. This is a type of multi-turn interaction. In our work, the number of turns of interactions between human and IQA models is similar to those between LEA and IQA models. We leave other forms of dialogues, potentially less well-defined and also lacking annotated public datasets, for future research.
>
> Table 6 accuracy results come from a non-interactive, Chain-of-thought enabled setting, and as a result, this setting already allowed the reasoning path to be fully fledged by the non-interactive model itself. Table 5 accuracy results are from LEA models after interacting with IQA models. Since both CoT outputs and interactions include reasoning processes, the difference between Table 5 and Table 6 naturally arises from the LEA-IQA models’ interaction process, suggesting that multi-turn interactions help LEA models reach higher accuracy scores.
>
> >2. The verification process for personas is lacking. There is no substantial evidence to consider that the performance differences by persona shown in Table 4 are reflective of the actual characteristics of those personas.
>
> We conduct a new experiment that shows the standard deviation in Table 4. The new table is in reply to question 2. Standard deviations show that personas affect evaluation performances.
>
> Additionally, we conduct an experiment demonstrating that personas impact LEA's performance. Using the "Expert" persona, we calculated the correlation between human evaluations and IQA-EVAL within a sub-group of pure experts. The results indicate that our personas accurately represent the pure expert group, as the correlation remains nearly consistent. In other words, if there is a dramatic change in correlation within the sub-group, it suggests that the proposed persona fails to represent the current group or the broader crowd.
>
> | 5-point Likert         | Helpfulness  |               |              |       | Fluency      |              |              |       |
> | ---------------------- | ------------ | ------------- | ------------ | ----- | ------------ | ------------ | ------------ | ----- |
> | LEA models             | TDA          | TB            | DA           | ρ     | TDA          | TB           | DA           | ρ     |
> | Human (Pure Expert)    | 4.69         | 4.00          | 3.73         |       | 4.36         | 3.96         | 3.26         |       |
> | IQA-EVAL (Pure Expert) | 4.37 | 3.57  | 3.33 | 0.778 | 4.20 | 3.40 | 2.97 | 0.786 |
> | Human                  | 4.60         | 3.84          | 3.52         |       | 4.35         | 3.84         | 3.22         |       |
> | IQA-EVAL (Expert)      | 4.17         | 3.08          | 3.12         | 0.756 | 4.47         | 3.84         | 3.40         | 0.787 |
> Based on the results, personas affect the performance of large language models.
>
> >3. A self-preference issue appears to have arisen in Table 5, and there is no control in place to address this.
>
> Please refer to the answer in the general author response on the top of the page.
>
> # Questions:
>
> >1. How does the Free Form Feedback in this study differ from the explanations provided by existing LLM-based evaluation methods (such as LLM-Eval and G-Eval)?
>
>
>
>
> The main difference is that the free-form feedback In this work evaluates interactions between LEA and IQA models, especially the performance of IQA models, while previous LLM-based automatic evaluation methods, like G-Eval, only evaluate direct outputs from Large Language Models (LLMs), which are non-interactive. Moreover, most related papers only provide numeric analysis, while we manually analyze feedback texts and correlate them to other results.
>
> >2. What is the total number of repetitions for the experiments mentioned in section 5.2? Could you provide the variance values for each experiment?
>
> We run the experiment in section 5.2 five times. The standard deviations are in the table below.
>
> |   |   |   |   |   |   |   |
> |---|---|---|---|---|---|---|
> |5-point Likert|Helpfulness|   |   |Fluency|   |   |
> |LEA models|TDA|TB|DA|TDA|TB|DA|
> |Human|4.60|3.84|3.52|4.35|3.84|3.22|
> |IQA-EVAL|4.30 (±0.06)|3.87  (±0.11)|3.93 (±0.13)|4.47 (±0.05)|3.67 (±0.08)|3.97 (±0.06)|
> |IQA-EVAL (Expert)|4.17 (±0.08)|3.08 (±0.09)|3.12 (±0.11)|4.47 (±0.02)|3.84 (±0.04)|3.40 (±0.04)|
> |IQA-EVAL (Critical-Thinker)|4.44  (±0.08)|4.02 (±0.13)|4.08 (±0.17)|4.64 (±0.06)|3.97 (±0.08)|4.10 (±0.08)|
> |IQA-EVAL (Adaptability-Seeker)|4.24 (±0.05)|3.67 (±0.11)|3.75 (±0.11)|4.52 (±0.08)|3.84 (±0.07)|3.884 (±0.09)|
> |IQA-EVAL (Clarity-Seeker)|4.45 (±0.07)|3.77 (±0.15)|3.80 (±0.12)|4.60 (±0.04)|3.85 (±0.04)|3.94 (±0.06)|
>
>
>
> >3. Why are the results for AmbigQA not included? Even if the performance of the IQA models is not optimal, the results should still be presented.
>
> The main reason for “-” in IQA-EVAL benchmarking (Table 5) is that weak IQA models cannot assist LEA models in answering questions at all. In most turns of interactions, these weak IQA models, for the task of outputing a disambiguated sentence, only repeat meaningless sentence pieces or questions proposed by LEAs. Thus, We use “-” instead of “0” in Table 5 to mark this inability of these models to complete the task.

---

> > ### Author Response · Authors · 2024-08-08
> > **More Responses to Weakness 1:**
> >
> > Both datasets, HotpotQA and AmbigQA, used in Table 5 are not multi-choice datasets. Their inputs and outputs are texts. HotpotQA requires LEA modes to output a short text including an answer to the given question and contexts. AmbigQA requires LEA models to disambiguous questions first and then answer that question by giving a short text.
> >
> > As you’ve suggested, we also benchmark IQA models in another dataset called Natural Questions [1]. This dataset comprises authentic questions posed by users about Wikipedia articles, demanding true multi-turn dialogues for resolution, akin to the setup in QuAC [2]. The experiment results are as follows:
> >
> > ----
> >
> > LEA: Claude-3
> >
> > | IQA models | Helpfulness | Fluency | # Queries | Accuracy |
> > | ---------- | ----------- | ------- | --------- | -------- |
> > | GPT3.5     | 4.86        | 4.88    | 2.82      | 0.42     |
> > | Claude     | 4.88        | 4.90    | 3.02      | 0.38     |
> > | Llama2     | 4.90        | 4.84    | 3.18      | 0.34     |
> > | Zephyr     | 4.84        | 4.90    | 3.02      | 0.28     |
> >
> > ----
> >
> > LEA: GPT-4
> >
> > | IQA models | Helpfulness | Fluency | # Queries | Accuracy |
> > | ---------- | ----------- | ------- | --------- | -------- |
> > | GPT3.5     | 4.12        | 5.00    | 2.76      | 0.44     |
> > | Claude     | 4.02        | 5.00    | 2.76      | 0.40     |
> > | Llama2     | 3.20        | 4.84    | 3.08      | 0.32     |
> > | Zephyr     | 3.30        | 4.86    | 2.92      | 0.36     |
> >
> > ----
> >
> > All numbers of queries in the first two tables are around 3, and each response to a query from IQA models contains an average of 2 sentences.
> >
> > Similar to Table 6 in the paper, we conduct non-interactive experiments on the following IQA models.
> >
> > | IQA models | # Sentences | Accuracy |
> > | ---------- | ----------- | -------- |
> > | GPT3.5     | 4.66        | 0.38     |
> > | Claude     | 3.16        | 0.34     |
> > | Llama2     | 5.21        | 0.30     |
> > | Zephyr     | 4.68        | 0.24     |
> >
> > Given the above number of sentences in each IQA model’s response, the non-interactive outputs are roughly equivalent to about two interaction turns, less than three turns in interactive outputs.
> >
> > Thus, **The interaction process of IQA-EVAL involves not only reasoning processes but also simulating genuine interactive multi-turn conversations. This suggests that the performances shown in the first two tables above are driven more by multi-turn interactions than by reasoning processes.** Furthermore, these **interactions lead to enhanced accuracy**, as demonstrated by the superior results in the first two tables compared to those in the last table (non-interactive).
> >
> > If you think we misunderstood your question and if you have further questions, please let us know.
> >
> > Reference:
> > 1. Kwiatkowski, Tom, et al. "Natural questions: a benchmark for question answering research." Transactions of the Association for Computational Linguistics 7 (2019): 453-466.
> > 2. Choi, Eunsol, et al. "QuAC: Question answering in context." arXiv preprint arXiv:1808.07036 (2018).

---

> > > ### Author Response · Authors · 2024-08-10
> > > **shall we have more discussions**
> > >
> > > Dear Reviewer,
> > >
> > > do you have any other questions of interest? In our previous rebuttal, we have added relevant explanations and several experiments, and we believe that these can to some extent address your concerns.
> > >
> > > We are eagerly looking forward to your response!

---

> > > > ### Author Response · Authors · 2024-08-12
> > > > **shall we have more discussions**
> > > >
> > > > Dear Reviewer,
> > > >
> > > > The discussion period is ending.
> > > >
> > > > Do you have any other questions of interest? In our previous rebuttal, we have added relevant explanations and several experiments, and we believe that these can to some extent address your concerns.
> > > >
> > > > We are eagerly looking forward to your response!

---

> > > > > ### Comment · Area_Chair_oCjR · 2024-08-12
> > > > > **Reviewer please respond**
> > > > >
> > > > > Dear reviewer,
> > > > >
> > > > > Thank you for your efforts in reviewing this paper. Now that the authors have provided their response, do you have any further comments?
> > > > >
> > > > > Thank you,
> > > > > AC

---

> > > > > ### Author Response · Authors · 2024-08-13
> > > > > **More discussions?**
> > > > >
> > > > > Dear Reviewer,
> > > > >
> > > > > The discussion period is ending.
> > > > >
> > > > > Do you have any other questions of interest? In our previous rebuttal, we have added relevant explanations and several experiments, and we believe that these can address your concerns.
> > > > >
> > > > > We are eagerly looking forward to your response!

---

### Author Rebuttal · Authors · 2024-08-07

We would like to thank all reviewers for their insightful reviews. Below we address common concerns by topics.

# Insufficient technical contribution, especially on the lack of prompt engineering.

Inspired by G-eval [4], we did conduct prompt engineering and designed our prompts by combining detailed instructions for different functions, such as task description, role description, metrics instructions, and evaluation instructions (we will add these information in the final version of the main paper). Each prompt is tuned based on a general question not included in IQA-Eval. We combine those instruction prompts flexibly based on evaluation stages.

We believe that the prompts in the paper are well-designed. To be detailed by our new experiments below, adding new tricks of prompting, such as using few-shot prompting or otherwise adding details and de-bias instruction does not show significant improvements.

# Concerns on LLM self-favoring bias

The impact of bias is low in our IQA-Eval.
During the develpoment of the prompts, we considered the impact of bias and manually evaluated all model interactions. Based on our evaluations, we tuned prompts as mentioned above to instruct our model to best reduce the impact of potential bias.

To prove that the bias effect is low, **we conduct a new experiment using a more complex prompt and few-shot, including effective debiasing prompts** (following FastChat [1] and Shaz et al [2]) to evaluate interactions. In the following table, the scores assigned by LEA models in response to our prompts align more closely with human evaluations.

||Helpfulness |||Fluency |||Accuracy |||
|----------------------------- |----------- |---- |---- |------- |---- |---- |-------- |---- |---- |
|LEA models |TDA |TB |DA |TDA |TB |DA |TDA |TB |DA |
|Human |4.60 |3.84 |3.52 |4.35 |3.84 |3.22 |0.69 |0.52 |0.48 |
|IQA-EVAL-GPT4 (our prompting) |3.67 |2.30 |2.10 |4.77 |3.87 |3.03 |0.87 |0.83 |0.67 |
|IQA-EVAL-GPT4 (New) |3.50 |2.23 |2.10 |4.40 |4.07 |3.53 |0.87 |0.83 |0.67 |

In our experiments on the MMLU dataset (Section 4&5), LEAs mainly evaluate the performance of IQA models in the interactions. Each LEA model evaluates all IQA models. **LEA models and IQA models are not the same models.** Thus, the impact of bias, as LLMs prefer themselves, is not a concern in the evaluation results presented in sections 4 (Meta Evaluation of IQA-EVAL Framework) and 5 (Effect of Assigning Persona to LEA).

Finally, in our benchmarking of different IQA models (Section 6). The impact of self-preference bias is low when using IQA-EVAL for benchmarking different IQA models. In IQA-EVAL benchmarking results (Table 5), line GPT3.5 appears to be the only one impacted by self-preference bias since it serves as both the LEA and IQA models. To prove that the self-preference bias effect is low, **we run the experiment using again the more complex prompt and few-shot like above**. The experiment results are highly similar to the line “GPT3.5” in Table 5, indicating that the self-preference bias issue has little impact on IQA-Eval.

 **HotpotQA**:

|LEA models |Helpfulness |Fluency |# Queries |Accuracy |
|-------------------------- |----------- |------- |--------- |-------- |
|IQA-EVAL-GPT3.5 (In paper) |4.72 |4.95 |1.49 |0.63 |
|IQA-EVAL-GPT3.5 (New) |4.68 |4.91 |1.35 |0.60 |
We will add these results and corresponding prompts to the paper.

----
[1] Zheng, Lianmin, et al. "Judging llm-as-a-judge with mt-bench and chatbot arena." Advances in Neural Information Processing Systems 36 (2024).

[2] Furniturewala, Shaz, et al. "Thinking Fair and Slow: On the Efficacy of Structured Prompts for Debiasing Language Models." arXiv preprint arXiv:2405.10431 (2024).

[3] Lee, Mina, et al. "Evaluating human-language model interaction." arXiv preprint arXiv:2212.09746 (2022).

[4] Liu, Yang, et al. "G-eval: Nlg evaluation using gpt-4 with better human alignment." arXiv preprint arXiv:2303.16634 (2023).

---

> ### Author Response · Authors · 2024-08-08
> **Additional global response: new methodology to further mitigate self-favoring bias**
>
> We designed a new methodology to further mitigate self-favoring bias. In this experiment, multiple LEA models evaluate the performance of IQA models during each interaction (“Multi-perspective”). In other words, we introduced third-party evaluations, where various LEA models assess the IQA models' performance instead of relying solely on the LEA model itself involved in the interaction. After evaluation, we use the average score from all LEA models as the final score. The results of IQA-Eval-Multi-perspective look as follows: (**Bold** numbers indicate they are the closest to human results.)
>
> |                                   | Helpfulness |          |          | Fluency  |          |          | # Queries |          |          | Accuracy |          |          |
> | --------------------------------- | ----------- | -------- | -------- | -------- | -------- | -------- | --------- | -------- | -------- | -------- | -------- | -------- |
> | LEA models                        | TDA         | TB       | DA       | TDA      | TB       | DA       | TDA       | TB       | DA       | TDA      | TB       | DA       |
> | Human                             | 4.60        | 3.84     | 3.52     | 4.35     | 3.84     | 3.22     | 1.78      | 2.57     | 2.66     | 0.69     | 0.52     | 0.48     |
> | IQA-EVAL-GPT4-Multi-perspective   | **4.32**    | **3.70** | **3.53** | 4.57     | **3.74** | 3.68     | **1.57**  | 2.27     | **2.3**7 | 0.87     | 0.83     | 0.67     |
> | IQA-EVAL-Claude-Multi-perspective | 3.96        | 3.13     | 3.10     | **4.29** | 3.51     | **3.22** | 2.20      | **2.67** | 2.07     | **0.6**7 | **0.53** | 0.57     |
> | IQA-EVAL-GPT3.5-Multi-perspective | 3.98        | 3.23     | 3.04     | **4.41** | 3.67     | 3.59     | **1.57**  | 1.77     | 2.00     | 0.63     | 0.47     | **0.53** |
>
> ----
>
> The correlations between IQA-Eval-Multi-perspective and human evaluations are as follows:
>
> |  | Helpfulness | Fluency   | Overall   |
> | ---------- | ----------- | --------- | --------- |
> | IQA-EVAL-GPT4-Multi-perspective | **0.702**   | 0.601     | **0.624** |
> | IQA-EVAL-Claude-Multi-perspective | 0.663       | **0.613** | 0.602     |
> | IQA-EVAL-GPT3.5-Multi-perspective | 0.641       | 0.552     | 0.533     |
>
> The overall result shows that GPT-4 is the best model aligning with human evaluations.

---

### Decision · Program_Chairs · 2024-09-25

**Decision:**

Accept (poster)

**Comment:**

The authors propose an interactive evaluation framework that uses an LLM to pose multi-turn questions to QA models under evaluation. The reviewers in general appreciate the usefulness of the framework, the fact that it is multi-turn and the results are correlated with human judgement. There are some concerns raised (including ethical concerns), such as the danger of self-bias during evaluation, the novelty of the work, and several technical details and questions. The authors conduct further experiments to support their rebuttal, addressing most concerns adequately.